# Air Quality during COVID-19 in Four Megacities: Lessons and Challenges for Public Health

**DOI:** 10.3390/ijerph17145067

**Published:** 2020-07-14

**Authors:** Patrick Connerton, João Vicente de Assunção, Regina Maura de Miranda, Anne Dorothée Slovic, Pedro José Pérez-Martínez, Helena Ribeiro

**Affiliations:** 1Global Health and Sustainability Doctorate Program, School of Public Health, University of São Paulo—USP, São Paulo 01246-904, Brazil; 2Department of Environmental Health, School of Public Health, University of São Paulo—USP, São Paulo 01246-904, Brazil; jianya@usp.br (J.V.d.A.); adslovic@usp.br (A.D.S.); lena@usp.br (H.R.); 3School of Arts, Sciences and Humanity, University of São Paulo—USP, São Paulo 03828-000, Brazil; remaura@usp.br; 4Department of Infrastructure and Environment, School of Civil Engineering, Architecture and Urban Design, University of Campinas—UNICAMP, Campinas 13083-889, Brazil; pjperez@unicamp.br

**Keywords:** air pollution, Covid-19, urban health, traffic reductions, activity restrictions

## Abstract

The study described in this manuscript analyzed the effects of quarantine and social distancing policies implemented due to the Coronavirus Disease 2019 (COVID-19) pandemic on air pollution levels in four western megacities: São Paulo in Brazil; Paris in France; and Los Angeles and New York in the United States. The study investigated the levels of four air pollutants—Carbon monoxide (CO), Ozone (O_3_), Fine Particulate (PM_2.5_) and Nitrogen dioxide (NO_2_)—during the month of March 2020, compared to 2015–2019, in the urban air of these metropolitan areas, controlling for meteorological variables. Results indicated reductions in the levels of PM_2.5_, CO and NO_2_, with reductions of the latter two showing statistical significance. In contrast, tropospheric ozone levels increased, except in Los Angeles. The beneficial health effects of cleaner air might also help prevent deaths caused by the epidemic of COVID-19 in megacities by diminishing pressure on hospitals and health equipment. Future actions for the re-starting of non-essential economic activities in these cities should take into consideration the overall importance of health for the individual, as well as for societies.

## 1. Introduction

The Coronavirus Disease 2019 (COVID-19) epidemic has had dramatic impacts on life around the globe, having spread to most countries, affecting over 9,300,000 people and causing over 478,000 deaths worldwide as of 23 June 2020 [1], and being compared by many to the 1918 flu pandemic. The economic impact of the virus has been significant as well, effectively halting many industries and sending financial markets into periods of rapid downturn. To help slow the spread of the virus, in addition to encouraging individual social distancing, governments have implemented shutdown, lockdown, stay-in-place, and even quarantine policies, with varying degrees of enforcement and effectiveness. These orders have drastically disrupted the routines of residents, altering environmental conditions in the affected areas as many commercial activities are halted and many employees are either experiencing reductions in hours or working remotely from their homes.

Media has been reporting the improvements in air quality in many cities around the world as a result of these policies and the closing down of non-essential activities. The benefits have been realized mainly in large global cities and their metropolitan areas. These circumstances thus provide a unique opportunity to assess the impact of anthropogenic activities that occur in urban areas on larger scales.

In large cities, air quality is influenced by pollutant emissions from vehicular and industrial sources. The main pollutants emitted and monitored in urban regions are nitrogen compounds (NO, NO_2_), carbon monoxide (CO), ozone (O_3_), sulfur dioxide (SO_2_), and fine and inhalable particulate matter (PM_2.5_, PM_10_). Large urban agglomerations suffer from problems related to pollution, congestion, lack of basic sanitation in peripheral (‘outskirt’) regions, homelessness, and unemployment, among other factors. These circumstances also make these areas more prone to the rapid transmission of diseases, since many people live in a small geographic region. The health crisis in these cities caused by COVID-19 is unprecedented, as are the demands being put on their healthcare systems.

According to the World Health Organization (WHO), close to 4 million deaths related to outdoor air pollution occur yearly around the world [2]. WHO recommends guideline values for each pollutant in order to protect health. Looking specifically at PM_2.5_, guideline values are 10 μg/m^3^ annual mean and 25 μg/m^3^ 24 h mean. However, in many cities those standards are frequently surpassed, with health implications for their citizens that include, but are not limited to, respiratory illnesses. A 2016 WHO study highlighted the impact of this exposure, using estimation of disease burden methods. In the study, percentages of populations exposed to PM_2.5_ were provided by country and by increment of 1 μg/m^3^; relative risks were then calculated for each PM_2.5_ increment, based on the integrated exposure-response functions (IER). Overall results of the study showed the following rate of deaths related to ambient air pollution: 14/100,000 inhabitants in Brazil; 8/100,000 inhabitants in France; and 7/100,000 inhabitants in the US [3].

Another study, published as a pre-print version, suggests that the majority of the pre-existing conditions that increase the risk of death for COVID-19 are the same diseases that are affected by long-term exposure to air pollution [4]. Based on this, the authors investigated the risk of COVID-19 deaths in the United States. As stated by the authors: “The results highlight that a small increase in long-term exposure to PM_2.5_ leads to a large increase in COVID-19 death rate, with the magnitude of increase 20 times that observed for PM_2.5_ and all-cause mortality. The study results underscore the importance of continuing to enforce existing air pollution regulations to protect human health both during and after the COVID-19 crisis” [4].

Furthermore, some scientists hypothesize that small particles in the air may serve to disseminate the virus among the population, based on the fact that there seems to be a positive correlation between the incidence of the viral infection cases and concentrations of particulate matter PM_10_ and PM_2.5_ in the air. These provide the basis and the vital conditions for the survival of the virus, in addition to transporting it in the atmosphere [5]. Isaifan [6] notes that chronic respiratory and cardiovascular diseases are well linked with air pollution as well as with COVID-19, which would imply that air pollution might be considered a secondary factor for these mortalities. Nevertheless, the author highlights that the virus’ inactivation in the particulate matter depends on climatic conditions. 

In another study that investigated these interactions, He, Pan, and Tanaka [7] studied the decrease in atmospheric pollution in Chinese cities during the epidemic and found that the counter-COVID-19 measures reduced PM_2.5_. Assuming these counter-virus measures, on average, lasted for 1 month (reducing PM_2.5_ by 8.40 μg/m^3^), and that the city lockdown further decreased PM_2.5_ in the affected cities by 13.9 μg/m^3^ (total 22.3 μg/m^3^), they estimated that the total number of premature deaths would be around 24,000 to 36,000. These numbers are significantly larger than the total number of deaths caused by COVID-19 in China and illustrate the enormous social costs associated with air pollution.

Containment measures were implemented in all four megacities considered in this study. The main sources of information about the policies implemented were official data from the Ministry of Health or Department of Health, and information released by state, metropolitan or city governments, as well as from press media. Appendix A shows descriptions of the measures taken by authorities in each megacity. The timeline of the confinement measures implemented by the four megacities is shown in Figure 1.

Under the scenario of anthropogenic restriction of activities due to the novel coronavirus (COVID-19), there is an increasing interest in determining the influence of current conditions, to compare to historical trends of urban air quality in large metropolitan areas. The prevailing conditions, human activities, and climatological parameters have a long-term influence on air quality and related human exposures and health effects, such as circulatory and respiratory diseases. 

In this context, this paper analyzes the impact of shutdown, confinement, or social-distancing orders in four megacities: the Municipality of São Paulo, Brazil (hereafter referred to as São Paulo); New York City (hereafter referred to as New York) and Los Angeles Metropolitan Area (hereafter referred to as Los Angeles) in the United States; and Paris, France. These cities adopted social confinement policies during the month of March 2020 as the number of infected inhabitants started to soar and hospital resources and intensive care units were/are at risk of collapse. Under these tragic circumstances, it is possible that reductions in air pollution levels might be helping to avoid respiratory infections and the use of hospital resources related to hospitalizations caused by air contaminants [6]. The study examines the impact of these measures on air quality, and their possible consequences for human health, based on monitoring data available for four common pollutants: carbon monoxide (CO), ground-level ozone (O_3_), nitrogen dioxide (NO_2_), and fine particulate matter (PM_2.5_), controlling for meteorological variables.

## 2. Materials and Methods 

This is an exploratory study that analyzed retrospective data on air quality and meteorology using statistical techniques, describing their correlation with actions undertaken to deal with the pandemic of the novel coronavirus in each of the four megacities.

### 2.1. Study Areas

The four megacities included in this study were chosen based on some overarching criteria; namely, they were heavily affected by the COVID-19 pandemic, the air pollution in each is very much related to motor vehicles and all have long histories in relation to air pollution monitoring and control. Additionally, each adopted social-confinement and quarantine policies during the month of March 2020, coinciding with when São Paulo started its social-distancing measures.

São Paulo was the reference area in this study; it was chosen based on being the residence for all the authors and for its historically bad air quality and administrative and legal actions taken to control atmospheric pollution. It is also an area where dozens of studies have been performed on the levels, origin, formation, and health effects of air pollution.

New York and Paris were also chosen as they were, along with São Paulo, the focus of a recent comparison of public policies for air pollution control [8]. Los Angeles was included in this investigation for its high dependency on motor vehicles and consequent air pollution. The four cities also have in common that they have been the epicenter of the pandemic within their respective regions. 

#### 2.1.1. São Paulo

The case of São Paulo was analyzed in a more detailed way. São Paulo (Appendix A) is one of the largest cities in the world. It covers an area of 1521 km^2^ with an estimated population of 12.2 million inhabitants and a density of 7570 inhabitants per km^2^. The Human Development Index (HDI) varies from high to very high and the portion of the population aged 60 and over is 14.46% [9].

In the past, the industries played an important role in the emission of pollutants but currently most of the atmospheric pollution comes from a fleet of about 8 million vehicles [10]. These emissions originate from a diverse vehicle fleet that includes light and heavy commercial trucks, cars, buses and motorcycles, which are fueled by a unique mix of fuels, mainly gasoline, ethanol and diesel, and some amount of liquefied petroleum gas (LPG). Contributions from motor vehicles account for 96.8% of total CO emissions, 75.3% of hydrocarbons (HC) emissions, and 63.9% of NOx emissions, while industry is the largest contributor of SO_X_ (83.3%) emissions, and also accounts for 36.1% of NOx emissions. Contributors to particulate matter PM_10_ include motor vehicles (40%), industry (10%), and the resuspension of soil and secondary aerosols (25% each) [10].

Carbon monoxide is no longer considered a problem pollutant, and sulfur dioxide levels agree with the WHO’s guidelines. Ozone, inhalable particulate (≤PM_10_), and fine particulate (PM_2.5_) continue to present problems for the city’s air quality (Appendix A). 

São Paulo experiences compromised air quality throughout the year, regularly exceeding air quality guidelines recommended by the WHO for both short and long-term levels of several air pollutants. It is estimated that air pollution in São Paulo is responsible for thousands of premature deaths annually, largely attributed to respiratory and cardiovascular effects of exposure to air pollution. It is important to note that the regional impacts of poor air quality are not experienced uniformly throughout the city, a result of the local impact of air pollution, as well as inequalities in access to health care and increased rates of respiratory cancer incidence and mortality within the poorest populations [11].

#### 2.1.2. New York

New York, combined with Newark, contains a population of approximately 8.5 million inhabitants [12]. Its five boroughs—Manhattan, Bronx, Queens, Brooklyn and Staten Island—cover an area of 789 km^2^, resulting in a population density of 10,800 inhabitants per km^2^. Although the city finds itself mostly in attainment with regards to US National Air Quality Standards (NAAQS), all counties are classified at ‘moderate’ for nonattainment of the 8-hr ozone standard, and Manhattan is classified as ‘moderate’ for nonattainment of the 1987 PM_10_ standard. More recent air quality control efforts in New York are linked to the launching of the Plan New York in 2007 by the New York City Mayor that established an action plan to tackle local environmental challenges such as air pollution and climate change. The same mayoral administration developed some innovative programs for the city, such as the closing of Times Square to vehicle circulation, and also the development of biking lanes over the entire city, an action that was one of the first of its kind in a metropolis.

However, the city still encounters air quality challenges, particularly due to emissions from water boilers, vehicular emissions during rush hours, and air pollution dispersion from surrounding cities (primary O_3_). The 2008–2018 New York City Community Air Survey revealed higher PM_2.5_ and NOx concentrations in Manhattan and Bronx. The report, from the New York City Dept. of Health and Mental Hygiene, shows lower values for air pollution in 2018 for PM_2.5_ and NO_2_ compared to a 2009 study, but highlights that air pollution levels are still high in areas with high traffic density, high population density, and boilers, and where industries still remain [13]. Ozone levels remain stable with health impact variations depending on season and geographic location [14].

#### 2.1.3. Los Angeles Metropolitan Area

The Los Angeles Metropolitan Area comprises Los Angeles and Orange counties. Los Angeles County is among the nation’s largest counties at 10,577 km^2^, and it has the largest population of any county in the nation with about 10.4 million residents in 2018, accounting for approximately 27 percent of California’s population, residing in 88 cities and approximately 140 unincorporated areas. It is an important industrial and financial center and is one of the most culturally and ethnically diverse communities in the world [15]. Los Angeles city is the most populous among all cities in Los Angeles County with about 4 million people [16]. Los Angeles County had a registered fleet of 8,154,560 motor vehicles on December 31, 2019, being 80.5% autos, 13.9% trucks, 3.6% trailers and 1.9% motorcycles, accounting for 22.8% of the California vehicle fleet [17].

Over the decades, the Los Angeles County–South Coast Air Basin has been in a state of nonattainment with respect to several National Ambient Air Quality Standards under 40 CFR Part 81, including: ozone 8 h (2004–present); carbon monoxide (1992–2006); lead (2010–present); nitrogen dioxide (1992–1997); PM_10_ (1992–2012); and PM_2.5_ (2005–present). While stationary sources contributed more to the makeup of air pollution in the past, today trucks and automobiles are responsible for the majority of emissions [18].

Orange County is included in the Los Angeles–Long Beach–Anaheim Metropolitan Statistical Area. The county has 34 incorporated cities, among them old cities like Santa Ana, Anaheim, and Fullerton. As of the 2018 census, the population was 3,185,968 inhabitants, with an area of 2460 km^2^ and a population density of 1295 inhabitants per km^2^. Orange County had a registered fleet of 2,943,942 motor vehicles on December 31, 2019, being 79.5% autos, 14.6% trucks, 3.8% trailers, and 2.1% motorcycles [12,17].

#### 2.1.4. Paris

Paris, the only European city included in this study, is also the smallest most densely populated, with 105.4 km^2^ and 21,289 inhabitants per km^2^. Paris is located in the Île de France region and concentrates the highest number of jobs in France among its 12 million inhabitants. The city center of Paris, also called “Paris intra-muros” (intramural Paris), has a population of 2.1 million.

Efforts to tackle air pollution are linked to the city administration of Mayor Delanoe, under which some unique actions were taken, such as the closing of the left bank of the river Seine—a former major corridor of vehicle circulation. Paris was one of the first cities to launch a bicycle sharing program, among other actions. The current Mayor, Anne Hidalgo, has also made air pollution control one of her primary efforts. This said, over the years, the city has experienced episodes of high levels of air pollution such as in March 2013 and 2014, when levels of particulate matter reached an average of 100 µg/m^3^ for three days in a row. These episodes have emphasized how air pollution continues to be a problem in Paris, despite previously implemented strategies. Deguen et al. [19] found that exposure to average NO_2_ concentrations for five years increased all causes of mortalities in the lower income section of the city, particularly when exposure to NO_2_ was above 55.8 µg/m^3^ [19].

The Parisian agency for air quality control (AIRPARIF), the local agency responsible for monitoring air quality in Paris, reported in its 2014 Air Quality Report that although there were improvements in the air pollution levels, they were insufficient, in particular for NO_X_ and PM near roads with intense traffic. It reported that 400,000 Greater Parisian inhabitants were exposed to daily levels above established standards [20]. More recent reports from the agency establish the traffic and residential sector (in great part due to residential heat) as primary sources of air pollution in Paris. Traffic corresponds to 69% of NO_x_ emissions, 36% of PM_10_ emissions, and 35% of PM_2.5_, while the residential sector represents 21% of NO_x_ emissions, 41% of PM_10_ emissions, and 49% of PM_2.5_ emissions [21]. Overall, the last ten years have seen a decrease of NO_x_ and PM emissions, primarily due to stricter and new vehicle emission restrictions and technologies [22] and cleaner sources of energy. However, regions with higher pollution levels remain linked to heavy traffic corridors and the use of diesel fuels.

### 2.2. Source of Data

Air quality data were obtained for CO, O_3_, NO_2_, and PM_2.5_ from local air monitoring agencies for the four cities. This study is primarily based on hourly monitoring data, with the exception of the analysis of O_3_ in Los Angeles, which is based on maximum 8-h averages. Additionally, the number of ozone exceedances were analyzed for São Paulo, based on the WHO guideline for this parameter (100 µg/m^3^, based on an 8 h average). For each city, all available monitoring data were compiled for each year. The hourly values were averaged for each day to obtain daily average concentrations for each pollutant and, when necessary, converted into units of either micrograms per cubic meter (µg/m^3^), used for PM_2.5_, NO_2_ and O_3_, or parts per million (ppm), used for CO.

For São Paulo, air quality data were obtained from state environment agency (CETESB) in its open online platform QUALAR (Cetesb, São Paulo, Brazil) [23]. CETESB has an extensive air quality monitoring network in São Paulo Metropolitan Area (Appendix A), with 30 air quality stations around the region, 17 of them in the city of São Paulo. For this study, hourly monitoring data were obtained for the Alto Tietê URGHI (Water Resources Management Unit) for all days in the month of March (2015–2020). No unit conversions were required for the São Paulo data. 

An additional analysis was also performed in São Paulo, using traffic data and air monitoring data for black carbon (BC) and CO_2_. In the Eastern Zone of São Paulo, BC and CO_2_ concentrations were monitored every 1 minute by an MAAP (Multiangle Absorption Photometer, Thermo Scientific, 5012) and CO_2_ Alphasense Non-Dispersive Infrared (NDIR) sensor, respectively. Although the data are only for a specific location in the city, it can be said that they represent well conditions in the city. Traffic data were provided by São Paulo Road Department (DER-SP) for five main São Paulo highways (Bandeirantes, Anhanguera, Castelo Branco, Raposo Tavares and Ayrton Senna). Aggregated records represent average daily traffic per direction divided by light duty vehicles (LDVs) and heavy-duty vehicles (HDVs), serving as proxies for the vehicle flows that access/egress the region.

For Paris, air quality data were obtained from the AIRPARIF Portail Open Data platform (AIRPARIF, Paris, France) [20,21]. This platform provides hourly data for all four pollutants considered in this study. The hourly values across all monitoring stations were averaged for each day to obtain daily average concentrations. This platform provides CO data in units of mg/m^3^, which were converted into units of ppm. 

For Los Angeles, monitoring data for O_3_ and PM_2.5_ were collected using the Air Data platform on the United States Environmental Protection Agency’s (US EPA) Daily Data Download platform (note: due to the short timeframe between the study period and data retrieval, the data for these pollutants available on the US EPA platform were likely sourced from the AirNow forecasting service). The Los Angeles-Long Beach-Anaheim core-based statistical area (CBSA) was used to obtain data for these pollutants on the US EPA platform. This CSBA includes both Los Angeles and Orange counties, and retrieves data from up to 16 monitors for these pollutants (although data availability was more limited for 2020 data); all available monitoring data for each year were included in the analysis. The US EPA platform provides O_3_ data in units of ppm, which was converted into units of µg/m^3^. Los Angeles monitoring data for NO_2_ were sourced from the South Coast Air Quality Monitoring District, and were derived from two monitoring stations: Central Los Angeles and North Orange County. The NO_2_ data were provided in units of parts per hundred million and converted into units of µg/m^3^. CO data for Los Angeles were sourced from the California Air Resources Board (CARB) Air Quality and Meteorological Information System (AQMIS), which provides daily averages for this pollutant from a network of 13 monitoring stations. The CO data retrieved were limited to the Los Angeles County monitoring stations.

All monitoring data for New York were retrieved from the New York State Department of Environmental Conservation Air Monitoring Website. Using this platform, data were exported for Region 2, which encompasses all 5 of the city’s counties. The platform provides hourly measurements, which were averaged to obtain daily averages. CO data from this platform are provided in units of ppb, which was converted into ppm; the NO_2_ data are provided in ppb and the O_3_ data are provided in units of ppm, both of which were converted into µg/m^3^.

In order to make a comparison and evaluate the influence of external parameters, data from the month of March of the previous five years (2015–2019) were compared to current data from March 2020. These included the first days of the month, when no restrictive measures were in place yet. A daily average of hourly readings was calculated for all days in the month of March for the previous 5 years and these values were compared to those for 2020.

Meteorological parameters including temperature, relative humidity, wind speed and precipitation were included in a statistical analysis. The meteorological data for São Paulo were also extracted from the QUALAR platform, with the exception of precipitation data, which were retrieved from National Institute of Meteorology (INMET) and from reports from the meteorological station of the Institute of Astronomy Geophysics and Atmospheric Science of University of São Paulo (IAG/USP). Meteorological data for New York and Los Angeles were obtained from the National Oceanic and Atmospheric Administration’s (NOAA) Local Climatological Data (LCD) tool. For Paris, meteorological data were obtained from Meteo France climatological data center.

### 2.3. Statistics

Daily air quality and climate data were collected for the four megacities. Measurements were obtained during six consecutive March months from 2015 to 2020. In total, 1302 atmospheric pollution and climate observations were collected, distributed over 186 days for each studied megacity. 

Basic descriptive statistics were compared and fit to a general linear model (GLM) using regional air quality network measurements, using CO, O_3_, NO_2_ and PM_2.5_ as the dependent/outcome variables and climatic long-term parameters—temperature, relative humidity and wind speed—as the explicative/independent/exposure variables. The model was adjusted for the COVID-19 scenario (year 2020) and years 2015–2019 (no activity constraint), to highlight the month of March 2020, with unusual restrictions of human activities and abnormal concentration of atmospheric pollutants. (Figure 1). The set of observations of contemporaneous measurements of pollutant concentrations and meteorology were used to demonstrate that ambient concentrations, in terms of CO and NO_2_, decreased as the main economic activities were mainly restricted from the cities. Although not included in the statistical analysis, precipitation data were also gathered to help explain changes in the pollutant concentrations.

The GLM estimated air quality at daytime *t*, March 2015–2020, using air quality samples from the mean regional network during March months when human activities such as traffic varied. The following regression model was used: (1)$$Air\_quality_{i,t\;\left(pollutant\;i,\;march\;day\;t\right)}=a_{0}+a_{1}P_{t}+W_{t}^{\prime }a_{2}^{w}$$
where *P_t_* is a dummy variable which ranges from 1 to 0 (1 for year 2020 and 0 for years 2015–2019), reflecting the March month related to human activity restrictions; *a*_0_ (intercept) and *a*_1_ (COVID-19 scenario effect) are regression coefficients obtained by ordinary least squares (OLS). Meteorological variables were also included to control for climate effects. Therefore, *W_t_*′ is a vector of climate records, which may impact atmospheric pollution and *a*_2_^*w*^ are the three regression coefficients related to: air temperature (°C), relative humidity (%) and wind speed (m/s).

Finally, the descriptive statistics of the two confinement periods were compared, scenarios outside of and within COVID-19, using observations and annual trends from 2015 to 2020 to check for significant differences of the means between groups: analysis of variance and Bonferroni post-hoc test.

## 3. Results and Discussion

### 3.1. Concentrations

Figure 2, Figure 3, Figure 4 and Figure 5 show daily averages of hourly monitoring values, compared to historical values for the comparison period (2015–2019) for all cities included in this study for the pollutants CO, NO_2_, O_3_ and PM_2.5_, respectively. Significant intervention dates are indicated in the graphs, and correspond to the weekday on which significant citywide shutdown orders (or equivalent) went into effect: São Paulo and New York—23 March; Los Angeles—19 March; Paris—17 March. It is possible to observe that in most cases, pollutant concentrations in 2020 were below the reference levels, with reductions at times appearing to coincide with the intervention date. It should be noted, however, that business closures, school shutdowns and work-from-home directives on smaller scales preceded the official shutdown orders in most cases and may have affected concentrations prior to their issuance. A notable exception to the reduction trend is O_3_, which increased in most cities (excluding Los Angeles) during 2020—a possible consequence of the reduction in NOx emissions, as there is a net decrease in available molecules for free oxygen atoms to bond to. A more detailed statistical analysis of the monitoring data will be presented. The concentration axes for the graphs change in accordance to the data range, to preserve their resolution, as average pollution levels vary among the cities. 

### 3.2. Meteorology

Table 1 shows the annual average and standard deviation of the meteorological parameters: temperature (T), relative humidity (RH), wind speed (WS) and cumulative precipitation (P) for all studied megacities. Meteorological conditions can have a significant influence on the concentration of pollutants. Changing from unfavorable to favorable conditions for the dispersion of pollutants normally occurs when a weather system reaches a region, making the atmosphere unstable and increasing winds. Precipitation also has a considerable influence. The occurrence of rainfall, in addition to being an indicator that the atmosphere is unstable, promotes pollutant removal. For example, unlike São Paulo, which experienced a very dry March 2020, Los Angeles had high levels of rainfall. Considering the four locations analyzed, from 2015 to 2020, the accumulated precipitation underwent major changes. The following statistical analysis will show the influence of temperature, relative humidity and wind speed on pollutant concentrations.

Appendix A shows the daily meteorological data for March 2020 in São Paulo. The temperature ranged from 20 to 26 °C and the relative humidity from 60% to 90%. Precipitation occurred at the beginning, middle and end of the month. From 23 March, when more restrictive measures started in São Paulo and until the 27th, the humidity decreased and the wind speed increased, contributing mainly to the removal of particulate matter. As will be seen below, despite the isolation measures, the concentration of PM_2.5_ decreased the least during this period. This is probably due to the fact that there are other sources of PM_2.5_ besides vehicles (the main cause of the reduction of the concentrations of the other pollutants analyzed) and also due to the formation of the secondary aerosol (mainly nitrate, sulfate and ammonium). From 21 March on, southeast winds, higher wind speeds and light precipitation could also have contributed to the dispersion of pollutants. However, statistical analysis showed that measures of social isolation had greater weight.

### 3.3. Statistical Analyses

Data were split into two periods, years 2015–2019 and 2020, corresponding to periods of no activity constraint and activity constraint caused by COVID-19 restrictions, respectively (Table 2a,b). Table 3 shows that air pollutants are improving at a higher rate than changes in meteorological parameters. The influence of human activities is more evident than meteorology; a statistical comparison of means shows more significant levels for air quality than for meteorological parameters (i.e., wind speed is not significant for all the cases).

Table 2a,b presents the mean, confidence interval (CI, level 95% obtained by perform bootstrapping using 1000 samples), number of measurements, and median, showing that during the period of restriction of economic activities and traffic reduction in March 2020, the mean values for the pollutants CO and NO_2_ were lower for the COVID-19 scenario than for the other scenario category for all the cities, based on the means, indicating that the transport of pollutants directly from anthropogenic sources such as light and heavy duty vehicle traffic affected air quality considerably. In March 2020, in São Paulo, the mean values for temperature and humidity were significantly lower in the COVID-19 period category. Similarly, in New York, temperature increased in March 2020. PM_2.5_ concentrations slightly decreased in the period of economic activity constraint, but at lower rates than CO and NO_2_, and no significant differences between scenario categories were observed. Sources other than vehicles probably contributed to secondary aerosol formation. In São Paulo, around 50% of PM_2.5_ is caused by secondary aerosols [10]. In general, in all four cities, air pollution improved considerably in March 2020 compared to the 2015–2019 average: CO decreased by 40% (0.23 ppm, with a CI of 0.20–0.26) in São Paulo, 24% (0.08 ppm, with a CI of 0.07–0.09) in Los Angeles, 19% (0.06 ppm, with a CI of 0.03–0.08) in New York and 67% (0.16 ppm, with a CI of 0.12–0.17) in Paris; NO_2_ decreased by 25% (7.99 µg/m^3^, with a CI of 5.19–10.58) in São Paulo, 38% (12.19 µg/m^3^, with a CI of 9.15–14.91) in Los Angeles, 24% (8.57 µg/m^3^, with a CI of 5.00–12.24) in New York and 39% (16.41 µg/m^3^, with a CI of 12.40–20.48) in Paris; PM_2.5_ decreased by 12% (1.70 µg/m^3^, with a CI of 0.01–3.36) in São Paulo, 37% (3.62 µg/m^3^, with a CI of 2.47–4.55) in Los Angeles, 24% (1.77 µg/m^3^, with a CI of 0.93–2.51) in New York and 28% (4.83 µg/m^3^, with a CI of 1.37–7.52) in Paris. Ozone concentrations increased: 30% (10.37 µg/m^3^, with a CI of 5.95–15.24) in São Paulo, 7% (3.99 µg/m^3^, with a CI 0.76–7.18) in New York, 12% (6.42 µg/m^3^, with a CI of 2.11–10.35) in Paris. Oppositely, in Los Angeles concentrations decreased by 10% (8.53 µg/m^3^, with a CI of 4.65–12.80). The absence of precursors, emitted by trucks and other vehicles, likely contributed to ozone formation [24]

The impact of activity restrictions under the COVID-19 scenario and meteorological variables were found to be significant in influencing air quality for all the cities and pollutants (except the effect of T in O_3_ formation in New York). In the generalized linear models, economic activity reductions tended to be more relevant for decreasing air pollution than climate conditions (Table 3). The models in São Paulo and Paris are better adjusted than for New York and Los Angeles (higher R^2^). Results confirmed the descriptive statistics, the analysis of variance and the post-hoc tests of the means (Table 2a,b).COVID-19 reduction effects related to CO are similar for all the cities, at ~0.10 ppm (95% confidence interval and standard errors were estimated), most likely due to the reduction of light duty vehicle traffic. São Paulo is the region where NO_2_ and PM_2.5_ improved least in absolute terms compared to the other cities: 4.00 µg/m^3^ vs. 11.01–14.18 µg/m^3^ and 0.54 µg/m^3^ vs. 2.03–2.99 µg/m^3^ for NO_2_ and PM_2.5_, respectively. Confinement measures in São Paulo were less restrictive compared to Los Angeles, New York and Paris. 

In the absence of comprehensive vehicle volume monitoring data for all four cities, mobility data made publicly available in direct response to the COVID-19 pandemic can serve as a proxy to give us some sense of how vehicle activity has been impacted by the COVID-19 restrictions. Such platforms include: Google COVID-19 Community Mobility Reports [25], which track the location of users who have opted to share their location history and uses median values of a five week period from 3 January–6 February 2020 as a baseline; Streetlight Data [26], which tracks smart phone and navigation device locations to calculate vehicle miles traveled (VMT) and uses average values for the month of January, 2020 as a baseline; and Apple Mobility Trends [27], which tracks Apple Maps directions requests and uses values from 13 January 2020 as a baseline. The data indicate that vehicle circulation volumes reduced significantly during the period analyzed. Specifically, the Google and Apple mobility data on transit stations shows reductions ranging from ~50 to ~75 percent (baseline values compared to values on selected intervention dates for each city). The Streetlight data, which are limited to the US, indicates VMT reductions ranging from ~55 to ~90 percent for counties in the New York and Los Angeles metropolitan areas. Lastly, the Apple mobility data indicate private vehicle reductions ranging from ~40 to ~80 percent.

In the case of Sao Paulo, these trends can be further supported by data from fuel sales and traffic flows at the main access/egress highways, showing [28,29] a ~20% reduction in gasoline and ethanol consumption and a ~13% reduction in diesel fuel consumption [28,29]. Additionally, gasoline/ethanol reductions in São Paulo were tied to light duty vehicles and private transport, whereas diesel reductions were mostly related to bus traffic decreases, given that diesel trucking continues the delivery of food and goods to shops and restaurants. Bus traffic represents 33% of CO_2_ diesel related emissions in São Paulo [30] and the passenger occupation of the vehicles increased during the pandemic episode [31].

Ozone formation was higher in New York and São Paulo, at 7.96–10.95 µg/m^3^. Using the ratio between the reduction of the pollutant in the coronavirus scenario (activity restriction coefficients in Table 3) and the average of the dependent variable (Total March 2015–2020 in Table 2a,b), it is possible to observe how important the restrictions were in reducing the pollutants and the effect of economic activities on atmospheric pollutants in São Paulo: 20% for CO, 13% for NO_2_ and 4% for PM_2.5_; LA: 30% for CO, 37% for NO_2_ and 33% for PM_2.5_; New York: 37% for CO, 41% for NO_2_ and 29% for PM_2.5_; Paris: 18% for CO, 32% for NO_2_ and 16% for PM_2.5_.

The GLM model was adjusted, showing that the effect of meteorology on air quality was significant for all the pollutants (Table 3). Regression coefficients related to meteorological parameters (temperature, relative humidity and wind speed) in São Paulo were higher than for the other three cities, showing the importance of seasonality in pollutant concentrations, given that March is the end of the rainy season (summer) and prevailing unstable conditions. For instance, an increase of 1 °C in temperature in São Paulo implied increases of 0.03 ppm (CO), 3.49 μg/m^3^ (O_3_), 1.18 μg/m^3^ (NO_2_) and 0.74 μg/m^3^ (PM_2.5_). The significant effect of a unit increase in wind speed on pollution reduction was also detected in São Paulo for CO (0.19 ppm), NO_2_ (10.40 μg/m^3^) and PM_2.5_ (5.49 μg/m^3^). In Los Angeles, an increase of 10% in humidity was related to a potential decrease of 3.30 μg/m^3^ of NO_2_. Similarly, in Paris, keeping the other parameters constant, a 1 m/s increase of wind speed was related to lower CO, NO_2_ and PM_10_ concentrations: 0.18 ppm, 18.53 μg/m^3^ and 9.91 μg/m^3^, respectively. Also, in Paris, a 10% reduction in humidity reduced PM_2.5_ concentrations by 1.80 ppm (similar to São Paulo). O_3_ formation was strongly related to humidity reduction in São Paulo (5.90 μg/m^3^ per 10% humidity decrease). 

Reductions in CO and NO_2_ and the opposite differences in O_3_ (in all the cities except LA) suggest that the improvement in air quality (in terms of CO and NO_2_) and O_3_ production might occur simultaneously with the reduction of economic activities due to COVID-19 (differences are statistically significant in Table 2a,b). From the results in Table 2a,b and performing paired-samples statistics post-hoc tests, we can conclude that the average reductions in the investigated cities, being 0.06–0.23 ppm for CO and 7.99–16.41 μg/m^3^ for NO_2_ during the 2015–2020 period, are not due to chance variation and can most probably be attributed to reductions in traffic and other urban activities (with a significance *p*-value for changes in CO and NO_2_ of less than 0.01). Similarly, the average increase in O_3_ of 10.37 μg/m^3^ in São Paulo can be related to lower NO emissions from vehicles (*p*–value < 0.01).

### 3.4. Other Results for Pollutants and Traffic in São Paulo

Figure 6a,b show the variations for each day in March 2020 of atmospheric pollutants compared to a baseline value for that day of the week. The baseline was built up using the median value, for the corresponding day of the week, during the March months of the previous five years (2015–2019): 0.55 ppm (CO), 32.98 μg/m^3^ (O_3_), 30.63 μg/m^3^ (NO_2_) and 13.60 μg/m^3^ (PM_2.5_). The graphs show an overall trend during March 2020 of CO and NO_2_ reductions and O_3_ formation, related to traffic flow reductions. The NO_2_ trend is confirmed by the data from the Tropomi Copernicus Sentinel-5P satellite (European Space Agency) and revealed a significant drop in NO_2_ concentrations, coinciding with strict mandatory quarantine measures in Sāo Paulo (Figure 6c–f). The satellite images show the evolution of the tropospheric vertical column of NO_2_ concentrations, at 10^−6^ mol/m^2^, from 10–13 March (2020, one week before the quarantine) compared to 24–27 March (2020, one week after the quarantine establishment). NO_2_ concentrations vary from day to day due to changes in climate and conclusions cannot be drawn based on just one day of data. By combining data from a specific time period, two weeks in this study, the meteorological variability is partially averaged and the impact of changes due to human activity starts to be seen. 

Because the chemistry in the atmosphere, especially in urban environments, is highly non-linear, the percentage drop in concentrations may differ considerably from the actual drop in emissions. However, the use of ground data in this study (meteorological and pollutant concentration records), helps to interpret the satellite concentrations, in order to estimate and confirm the influence of the containment measures. There is no clear trend regarding PM_2.5_ concentrations. Increases in wind speed which might increase dust resuspension from streets, secondary aerosol formation and transport from other regions, together with decreases in temperature and humidity, may have offset the direct PM_2.5_ mitigation from anthropogenic emissions. The small reductions in PM_2.5_ may not be as significant as those in CO and NO_2_: the p-value being greater than 0.10 for reductions in concentration levels in the GLM shows that the COVID-19 restriction did not significantly reduce anthropogenic related PM_2.5_ levels (unlike CO and NO_2_ levels).

Figure 7 shows the main atmospheric pollutants (CO, CO_2_ and BC) emitted by mobile sources and traffic volume. According to Figure 7, traffic decreased around 40% between the 19th of March (red reference line) and the end of the month, for both light and heavy vehicles. On 20 March, a state of emergency was declared in São Paulo (Figure 1). The vehicle traffic reduction coincided with decreases of pollutants: BC and CO decreased by 63.6% and 53.6%, respectively. The ratio between CO_2_ and CO concentrations (traced green line) can be used to estimate the contribution of fossil fuel combustion due to anthropogenic activities [32]. This ratio considerably increased after March 19 (red line in the figure), indicating the importance of vehicle reductions compared to other biological sources. CO_2_ decreased by 0.2% (0.7 ppm) during this period, almost half of the growth rate (~2 ppm/year) of the accumulation of CO_2_ concentrations in the atmosphere [33].

## 4. Potential Benefits and Other Impacts

### 4.1. Air Pollution, COVID-19 and Potential Health Impacts

Air pollution has been identified as a factor that can exacerbate the impact of and potentially serve to disseminate the COVID-19 virus [4,5,6]. Although health data for the month of March 2020 from the four megacities are not yet fully available and thus could not be included in the analysis, we can hypothesize on the possible positive health impacts of the social containment measures based on the literature and on inferences taking into consideration the results of this study regarding air quality and meteorological data. Our point is that a minimization of short-term health effects caused by reduced pollution levels could diminish pressure on the demand for health equipment and intensive care units, which then could be used for COVID-19 patients. A comprehensive accounting of the health impacts of the restriction measures is uncertain and demands future studies.

#### 4.1.1. Carbon Monoxide—CO

Carbon monoxide was used in this study mainly as a proxy for the amount of light-duty vehicle traffic. Concentrations have been very low in São Paulo, as shown in Appendix A, with no exceedance of the WHO guidelines (8 ppm, 8-h average) in the last 10 years. Thus, the reduction that occurred in the four cities (Table 2a,b), in March 2020, in relation to the same period in 2015–2019, may be assumed to have little relevance regarding beneficial health effects. 

#### 4.1.2. Nitrogen Dioxide—NO_2_

Children, elderly and asthmatic people are in the susceptible group for NO_2_ pollution, which might increase allergic inflammation and airway responsiveness in adults with asthma in short term exposures to this pollutant [34].

Our results show daily mean values for NO_2_ of 23.6 μg/m^3^, 19.7 μg/m^3^, 27.3 μg/m^3^, and 25.4 μg/m^3^ in March 2020 and a reduction of 8.0 μg/m^3^, 12.2 μg/m^3^, 8.6 μg/m^3^, and 16.4 μg/m^3^ in relation to average values for March in period 2015–2019 for São Paulo, Los Angeles, New York and Paris, respectively. In spite of the reduction observed during confinement, concentrations are still around the lower limit observed by the US Environmental Protection Agency [34], possibly increasing respiratory diseases, especially chronic obstructive pulmonary diseases, exacerbating asthma, and increasing emergency visits. 

Although it is a very new disease, the literature already includes a few studies aimed at the association of air pollution with COVID-19 incidence. One of them was done in China by Zhu et al. [35]. The authors advise caution in interpreting the results, since the causal relationship was not yet clearly established. In relation to NO_2_, their results showed that an increase in the NO_2_ concentration was positively associated with daily counts of COVID-19 cases. 

Thus, the reduction achieved by confinement measures in the four megacities possibly prevented the occurrence of respiratory diseases and may have helped to reduce hospital admissions, thus reducing the pressure on hospital and medical services and infrastructure, and possibly diminishing the burden of cases and deaths by COVID-19 in the four cities, with lesser intensity in São Paulo and New York.

#### 4.1.3. Fine Particulate Matter—PM_2.5_

Fine particulates showed decreases in their concentrations during the studied period when COVID-19 confinement was in place, compared to the reference period used in this study. São Paulo had the lowest reduction (12%), and Los Angeles the highest (37%), followed by Paris (28%) and New York (24%). Based on WHO [3] data on the effects of fine particles on deaths from the four countries involved in the study, if we apply the formula indicated of 1 μg/m^3^ reduction for this short period, we hypothesize that, if these lower levels persist, in one year around 500 premature deaths could be avoided for São Paulo and Paris, around 220 premature deaths avoided for New York, and around 340 premature deaths avoided for Los Angeles. On the other hand, results from the study undertaken by Wu et al. [4] with data from 3000 counties in the United States, adjusted for population size, hospital beds, number of individuals tested, weather, socioeconomic and behavioral variables, including obesity and smoking, showed a robust statistical association of air pollution and deaths by COVID-19. 

Regarding COVID-19 incidence, Zhu et al. [35] found a positive association with PM_2.5_ concentration in Chinese cities. If we infer that the same applies to the four cities analyzed in our study, the confinement measures might represent additional benefits. 

#### 4.1.4. Tropospheric Ozone—O_3_

Ozone is a powerful oxidant and may cause reductions in lung function and increases in hospital admission rates. Other health effects, such as respiratory symptoms or short-term changes in mortality rates could also result from ozone exposure. The WHO [36] considered 100 μg/m^3^, 8-h average, as a limit for exposure. A very recent impact statement assessment by the US EPA [37] showed that epidemiologic studies are consistent with an association of lung function impairment at O_3_ concentrations as low as 33 ppb (62.2 μg/m^3^, 8-h average). 

Daily mean ozone concentrations, in March 2020, ranged from 45.0 μg/m^3^ in São Paulo to 80.2 μg/m^3^ in Los Angeles, and exceedances of the WHO guidelines (100 μg/m^3^, 8-h average) in São Paulo were much higher in the month of March 2020 than during the 5 previous years (Appendix A). In contrast with the other three pollutants, O_3_ mean concentrations were higher in March 2020 in São Paulo, New York, and Paris, resulting in possible increased risks to health. In Los Angeles, average ozone concentrations decreased, resulting in a possible health benefit in this city. According to Table 1, in March 2020, precipitation levels in Los Angeles were much higher than in the previous 5 years, which may have contributed to the drop in ozone concentrations.

### 4.2. Circulation Restrictions and Air Quality

This research has combined air pollution and meteorological data in four of the world’s most renowned cities. It showed how interventions related to COVID-19 prevention influenced air pollution levels. The measures undertaken within the four cities to close educational, cultural, leisure, and commercial activities and switch to teleworking had a direct impact on the number of vehicles circulating, and on air quality. These results are similar to other cities where lockdown measures reduced levels of NO_2_, BC, and, to a lesser extent, PM_10_ [38,39]. Reductions in the number of vehicles as a result of interventions aimed at reducing contagion also provide a unique opportunity for cities to re-think mobility in the long-term.

City officers, as they “unlock” their cities, will have to take into account mobility issues as a central point for containing the spread of the disease; how people transit will become a central consideration. This is accentuated by preliminary results pointing out that air pollution levels could act to exacerbate COVID-19 transmission and worsen health effects, therefore serving as an indicator of vulnerable zones [40,41,42]. This vulnerability enhances the importance for New York, Paris, Los Angeles, and São Paulo to foster integrated environmental health approaches when developing interventions in times of health crises and in general. Transport interventions are key to increasing accessibility and lowering segregation [43], and also to reducing air pollution. COVID-19 represents a unique opportunity to switch habits to more sustainable alternatives and include safety measures for those who rely on public transport. This includes more awareness of the central role transport systems have in urban centers. Re-thinking commuting in the deconfinement will also include consideration of rapid transit lanes, aerial trams, and active transports that have been found to reduce commute times and enhance health [44]. 

#### 4.2.1. New Opportunities for Better Air Quality and Health

Other aspects to be considered are the health and economic benefits of active forms of mobility [45]. Furthermore, during the pandemic, the WHO has recommended riding bicycles and walking for those that need to commute to ensure social distancing and provide physical activity [46]. New York and Paris have temporarily made space for bicycle lanes during the COVID-19 pandemic and are planning on continuing as cities slowly re-open. Other cities like Barcelona, Berlin, Bogotá, Boston, and Montreal have already implemented measures to prioritize the use of public spaces for active modes of transit [47]. In São Paulo there are no updates on such initiatives, however it is important to stress that progress has been made over the last decade in promoting active transport modes and public transport to decrease private vehicle use. 

In an immediate sense, the complex relationships between meteorological variables, air pollution levels and the dissemination of COVID-19 should be taken into consideration by city managers as they begin to relax containment measures. A recent study looked at the influence of weather on the spread of COVID-19, finding that variables such as humidity, temperature, and solar radiation can influence the proliferation and impact of the virus, with colder temperatures in particular being linked to higher mortality rates [48]. Although more research is clearly needed to better understand these relationships, this dynamic is of particular concern for São Paulo as the city heads into winter, a period that is associated with elevated levels of PM pollution. 

#### 4.2.2. Reducing Work Related Trips to Improve Air Quality and Save Lives

In addition to re-evaluating how we commute, Musselwhite et al. [49] stress the role of transport on what they call hypermobility and how it fosters the spread of disease. A solution proposed by the writers is a reduction of speed in our mobility and a switch to a more locally focused life where social life is neighborhood-centered. In many ways, the lockdown and social distancing measures have shown that there is potential for some industries to switch to teleworking, which in the short-term has an impact on the spread of COVID-19, but will also positively impact air pollution levels in the long-term. Giovanis [50] found in a study in Switzerland that teleworking had the potential to reduce traffic volumes by 2.7 percent and ameliorate air quality. This said, it is important to highlight that teleworking also implies access to technology, which in middle- and low-income countries is far from guaranteed. Low skill workers and the majority of students will still have to rely on public transport commutes, highlighting the importance of improving transportation for all and focusing on active transport as an environmental health priority. Of course, the long-term improvements highlighted above presume a post-pandemic scenario; the worst future externality would be that residents opt to use individual vehicles to protect themselves from contamination by the virus, both during and after the COVID-19 crisis, annulling decades of efforts to regulate air quality in megacities. 

The findings and their implications should be discussed in the broadest context possible. Future research directions may also be highlighted.

## 5. Conclusions

During March 2020, human activities decreased due to confinement of urban populations, leading to an improvement of atmospheric conditions in all of the four cities studied. Compared to the previous five years, pollutant concentration reductions were more evident for CO and NO_2_: CO decreased by 40% in São Paulo, 24% in Los Angeles, 19% in New York and 67% in Paris, while NO_2_ decreased by 25% in São Paulo, 38% in Los Angeles, 24% in New York and 39% in Paris. PM_2.5_ decreased by 12% in São Paulo, 37% in Los Angeles, 24% in New York and 28% in Paris. Ozone concentrations increased in São Paulo, New York, and Paris. As an exception to this trend, O_3_ concentrations decreased in Los Angeles. 

The effects of meteorological parameters were also considered, showing that air pollutants were improved at a higher rate through the restrictions when compared to changes in meteorological parameters; this means that the effect of the restrictions imposed and, consequently, the reduction of vehicle traffic, was more important in improving air quality than the effects of meteorology. 

NO_2_ and PM_2.5_ improved the least in absolute terms in São Paulo compared to the other cities. The fact that the reduction of the pollutant CO was similar in all cities shows that the margin for the reduction of this pollutant is smaller, and that vehicle technology for light cars is well developed. The main sources and composition of the PM_2.5_ need to be better studied, since it is one of the most harmful pollutants for human health. Between the 19th of March and the end of the month, traffic decreased around 40% for light and heavy vehicles in São Paulo (CO_2_ decreased by 0.2%, ~0.7 ppm) and the ratio of CO_2_ to CO concentrations increased, indicating the importance of vehicle reductions compared to other biological sources.

Restriction policies certainly had a strong influence on reducing the concentration of pollutants, but the social reality of each location contributes to the degree of adherence to restrictions. 

Given the results of this study, it can be argued that social containment is not only avoiding infection and deaths by COVID-19 by restricting crowds in the streets and diminishing the risk of community transmission, but is also avoiding diseases of the respiratory tract and cardiovascular system by reducing concentrations of air pollutants. Air quality improvements and decreases of cardiovascular and respiratory diseases may have offset some of the negative effects of COVID-19 during the month of March. The results of this paper highlight the importance of public policy and measures to protect health for individuals and society as a whole.

## Figures and Tables

**Figure 1 ijerph-17-05067-f001:**
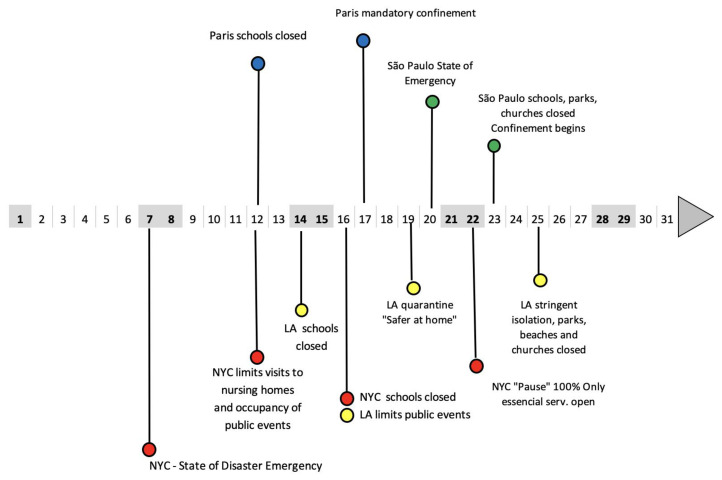
Timeline of the confinement measures taken in Sao Paulo, New York, Paris, and Los Angeles during March 2020 to prevent infection by the novel coronavirus.

**Figure 2 ijerph-17-05067-f002:**
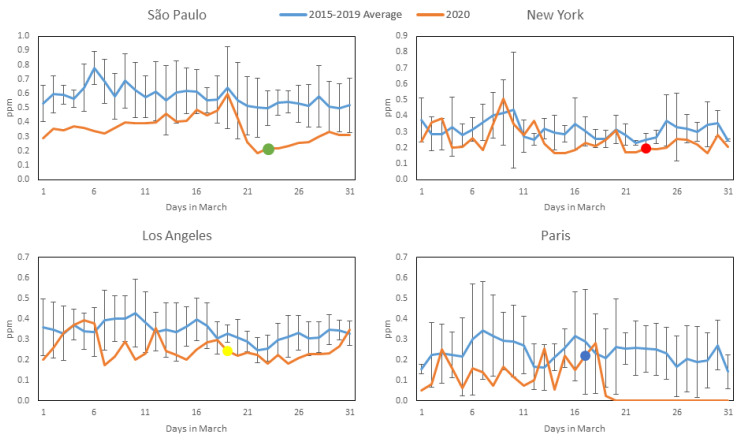
CO concentration profiles during March 2020 (Orange) compared to average and standard deviation of March concentrations over the previous 5 years (Blue) for the four studied cities (most significant COVID-19 intervention date indicated with colored dot). All available station data used for each city (see Section 2.2).

**Figure 3 ijerph-17-05067-f003:**
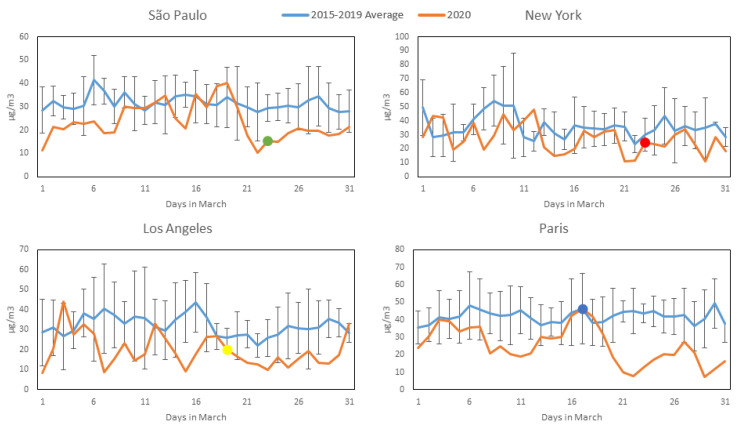
NO_2_ concentration profiles during March 2020 (Orange) compared to average and standard deviation of March concentrations over the previous 5 years (Blue) for the four studied cities (most significant COVID-19 intervention date indicated with colored dot). All available station data used for each city (see Section 2.2).

**Figure 4 ijerph-17-05067-f004:**
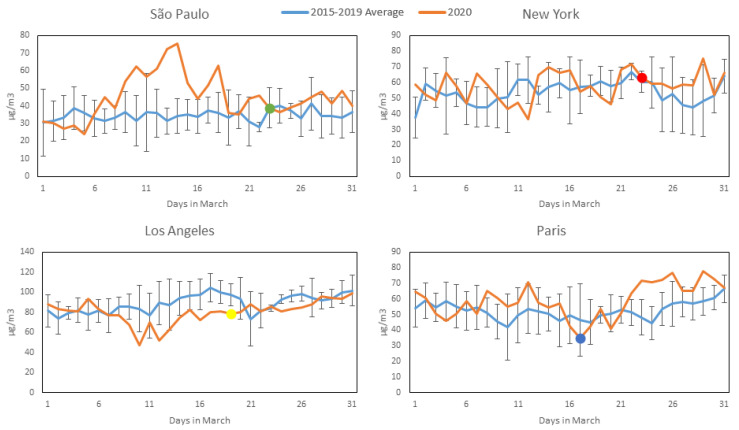
O_3_ concentration profiles during March 2020 (Orange) compared to average and standard deviation of March concentrations over the previous 5 years (Blue) for the four studied cities (most significant COVID-19 intervention date indicated with colored dot). All available station data used for each city (see Section 2.2).

**Figure 5 ijerph-17-05067-f005:**
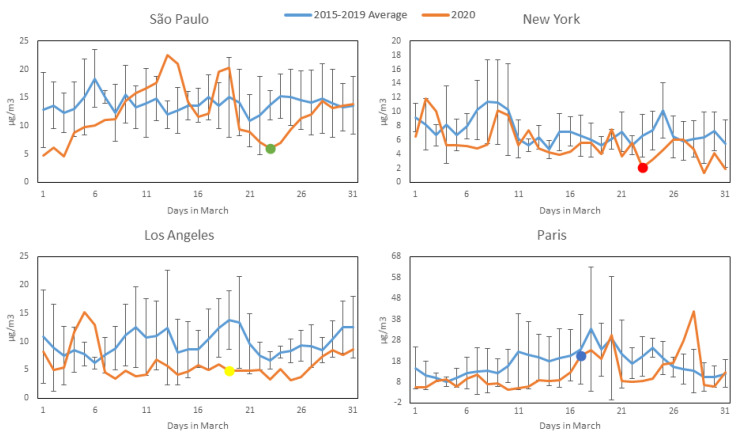
PM_2.5_ concentration profiles during March 2020 (Orange) compared to average and standard deviation of March concentrations over the previous 5 years (Blue) for the four studied cities (most significant COVID-19 intervention date indicated with colored dot). All available station data used for each city (see Section 2.2).

**Figure 6 ijerph-17-05067-f006:**
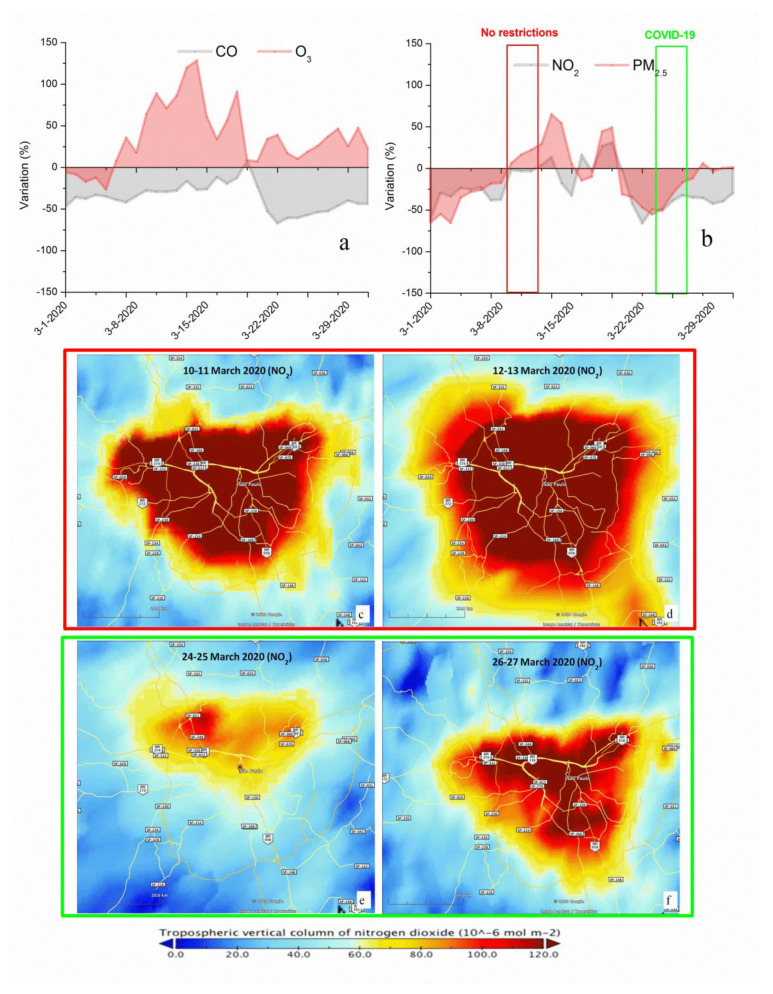
Variations for each March day of 2020 compared to a baseline value, median of March concentration levels of years 2015–2019 for São Paulo MA, (**a**) CO and O_3_, (**b**) NO_2_ and PM_2.5_. Highlighted periods in red and green correspond to maps c, d and e, f, respectively. Maps of NO_2_ columns elaborated from satellite data from the European Space Agency, Tropomi Copérnico Sentinel 5P. (**c**,**d**) represent a period before the confinement. (**e**,**f**) correspond to the period of activity restriction (COVID-19 scenario).

**Figure 7 ijerph-17-05067-f007:**
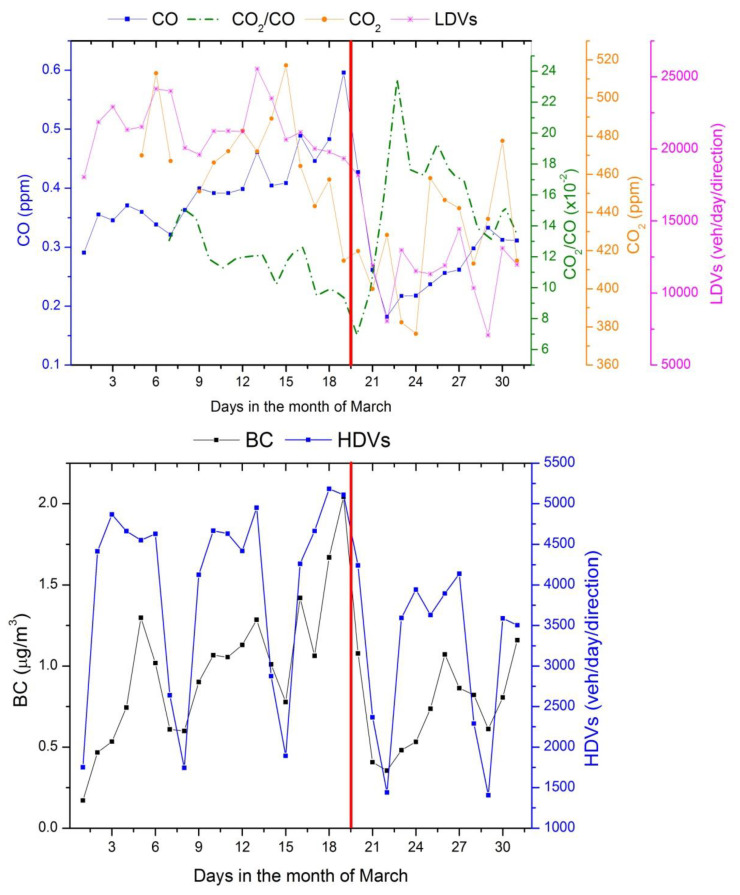
CO, CO_2_ and black carbon (BC) concentrations related to traffic volume in São Paulo.

**Table 1 ijerph-17-05067-t001:** Annual average and standard deviation for meteorological parameters for the month of March from 2015 to 2020.

**São Paulo**					**Los Angeles**			
March of	T (°C)	RH (%)	WS (m/s)	P (mm)	T (°C)	RH (%)	WS (m/s)	P (mm)
2015	22.7 ± 2.8	79 ± 13	1.6 ± 0.6	204.2	18.7 ± 3.4	58 ± 19	1.9 ± 0.3	23.0
2016	23.7 ± 3.0	77 ± 13	1.7 ± 0.7	198.3	16.2 ± 1.8	67 ± 13	2.4 ± 0.9	60.6
2017	23.2 ± 2.7	74 ± 12	1.9 ± 0.6	141.4	16.9 ± 1.8	62 ± 17	2.2 ± 0.8	3.8
2018	24.4 ± 3.3	75 ± 15	1.5 ± 0.6	222.2	14.9 ± 1.8	67 ± 20	2.3 ± 0.9	95.6
2019	23.4 ± 3.3	76 ± 14	1.7 ± 0.6	329.2	15.6 ± 2.4	62 ± 18	2.4 ± 0.8	86.5
2020	22.3 ± 3.6	72 ± 14	1.9 ± 0.6	69.6	15.4 ± 1.8	62 ± 13	2.3 ± 0.6	162.7
**New York**					**Paris**			
March of	T (°C)	RH (%)	WS (m/s)	P (mm)	T (°C)	RH (%)	WS (m/s)	P (mm)
2015	3.1 ± 3.8	57 ± 18	4.4 ± 1.6	197.6	0.95 ± 0.56	69 ± 9	1.0 ± 0.6	32.5
2016	8.7 ± 4.6	57 ± 13	4.5 ± 1.7	48.3	1.38 ± 0.43	71 ± 8	1.4 ± 0.4	89.3
2017	4.1 ± 5.3	54 ± 21	5.6 ± 1.9	204.9	1.25 ± 0.41	74 ± 9	1.4 ± 0.4	67.8
2018	4.2 ± 2.5	60 ± 16	5.6 ± 1.9	193.0	1.17 ± 0.30	75 ± 7	1.2 ± 0.3	81.8
2019	4.8 ± 4.5	57 ± 18	5.0 ± 1.8	146.2	1.52 ± 0.63	70 ± 7	1.5 ± 0.6	43.9
2020	8.2 ± 3.2	61 ± 19	4.7 ± 1.2	165.9	1.45 ± 0.52	68 ± 15	1.5 ± 0.5	50.3

**Table ijerph-17-05067-t002a:** (**a**)

São Paulo	Mean (CI)	N	Median	Los Angeles	Mean (CI)	N	Median
**a (March 2015–2019) ^1^**				**a (March 2015–2019) ^1^**			
**Pollutant concentrations**				**Pollutant concentrations**			
CO (ppm)	0.58 (0.55–0.61)	155	0.55	CO (ppm)	0.34 (0.31–0.34)	155	0.33
O_3_ (µg/m^3^)	34.67 (33.05–36.39)	155	32.98	O_3_ (µg/m^3^)	88.11 (85.46–90.76)	155	88.87
NO_2_ (µg/m^3^)	31.58 (30.10–33.09)	155	30.63	NO_2_ (µg/m^3^)	31.86 (29.74–33.98)	142	28.71
PM_2.5_ (µg/m^3^)	13.91 (13.16–14.63)	155	13.60	PM_2.5_ (µg/m^3^)	9.51 (8.79–10.36)	155	8.55
**Meteorological Factors**				**Meteorological Factors**			
T (°C)	23.45 (23.19–23.72)	155	23.53	T (°C)	16.42 (16.01–16.88)	155	16.2
RH (%)	76.36 (75.24–77.57)	155	76.54	RH (%)	64.13 (61.47–67.02)	155	68.30
WS (m/s)	1.71 (1.64–1.78)	155	1.66	WS (m/s)	2.26 (2.14–2.39)	155	2.00
**b (March 2020) ^1^**				**b (March 2020) ^1^**			
**Pollutant concentrations**				**Pollutant concentrations**			
CO (ppm)	0.35 (0.32–0.39)	31	0.36	CO (ppm)	0.26 (0.23–0.28)	31	0.23
O_3_ (µg/m^3^)	45.03 (40.25–49.61)	31	44.14	O_3_ (µg/m^3^)	80.22 (75.98–84.22)	31	80.79
NO_2_ (µg/m^3^)	23.59 (21.15–26.25)	31	21.46	NO_2_ (µg/m^3^)	19.67 (16.76–22.81)	31	17.72
PM_2.5_ (µg/m^3^)	12.21 (10.55–13.94)	31	11.59	PM_2.5_ (µg/m^3^)	6.12 (5.12–7.26)	31	5.04
**Meteorological Factors**				**Meteorological Factors**			
T (°C)	22.39 (21.75–23.00)	31	22.62	T (°C)	15.38 (14.75–16.05)	31	14.90
RH (%)	71.84 (69.79–74.00)	31	70.37	RH (%)	62.25 (57.92–66.53)	31	63.40
WS (m/s)	1.93 (1.78–2.09)	31	1.817	WS (m/s)	2.31 (2.09–2.53)	31	2.30
**Total (March 2015–2020)**				**Total (March 2015–2020)**			
**Pollutant concentrations**				**Pollutant concentrations**			
CO (ppm) ^2,^**	0.54 (0.51–0.57)	186	0.52	CO (ppm) ^2,^**	0.32 (0.30–0.33)	173	0.32
O_3_ (µg/m^3^) ^2^**	36.40 (34.79–38.15)	186	34.31	O_3_ (µg/m^3^) ^3,^*	86.70 (84.33–89.09)	186	86.89
NO_2_ (µg/m^3^) ^2,^**	30.25 (28.94–31.61)	186	29.47	NO_2_ (µg/m^3^) ^2,^**	29.68 (27.69–31.73)	173	26.85
PM_2.5_ (µg/m^3^) ^3^	13.62 (12.95–14.34)	186	13.31	PM_2.5_ (µg/m^3^) ^3,^**	8.90 (8.24–9.64)	186	7.91
**Meteorological Factors**				**Meteorological Factors**			
T (°C) ^3,^*	23.27 (23.01–23.51)	186	23.33	T (°C) ^3^	16.24 (15.86–16.62)	186	16.00
RH (%) ^3,^*	75.61 (74.59–76.59)	186	75.61	RH (%)	63.79 (61.38–66.19)	186	66.40
WS (m/s) ^3^	1.74 (1.67–1.81)	186	1.68	WS (m/s)	2.27 (2.15–2.39)	186	2.00

Notes: ^1^ a—no confinement (2015–2019) and b—confinement COVID-19 scenario (2020); ^2^ There are significantly distinct sets of means between years without march activity restriction (a, 2015–2019) and 2020 (b, coronavirus scenario), significantly different α = 0.05, Bonferroni test; ^3^ There are significant differences between some pairs of means and years, but there is no significantly distinct sets of means, a vs. b; ** analysis of variance *p* < 0.001, * analysis of variance *p* < 0.01, *p*–value represents results of the ANOVA procedure and compares the two groups of years, with (2020) and without (2015–2019) COVID-19 scenario.

**Table ijerph-17-05067-t002b:** (**b**)

New York	Mean (CI)	N	Median	Paris	Mean (CI)	N	Median
**a (March 2015–2019) ^1^**				**a (March 2015–2019) ^1^**			
**Pollutant concentrations**				**Pollutant concentrations**			
CO (ppm)	0.31 (0.29–0.33)	155	0.27	CO (ppm)	0.24 (0.21–0.26)	155	0.23
O_3_ (µg/m^3^)	54.01 (51.72–56.30)	155	57.29	O_3_ (µg/m^3^)	52.50 (50.29–54.51)	155	53.82
NO_2_ (µg/m^3^)	35.88 (33.28–38.51)	155	33.59	NO_2_ (µg/m^3^)	41.80 (39.82–43.79)	155	42.01
PM_2.5_ (µg/m^3^)	7.26 (6.71–7.80)	155	6.14	PM_2.5_ (µg/m^3^)	17.15 (15.26–19.30)	155	12.25
**Meteorological Factors**				**Meteorological Factors**			
T (°C)	4.98 (4.28–5.70)	155	4.50	T (°C)	6.79 (6.10–7.51)	155	7.66
RH (%)	56.94 (53.94–59.83)	155	55.40	RH (%)	71.81 (70.47–73.13)	155	71.80
WS (m/s)	5.05 (4.79–5.34)	155	4.80	WS (m/s)	1.25 (1.18–1.33)	155	1.21
**b (March 2020) ^1^**				**b (March 2020) ^1^**			
**Pollutant Concentrations**				**Pollutant concentrations**			
CO (ppm)	0.25 (0.22–0.28)	31	0.22	CO (ppm)	0.08 (0.05–0.12)	31	0.06
O_3_ (µg/m^3^)	58.01 (54.99–61.13)	31	58.60	O_3_ (µg/m^3^)	58.92 (54.81–62.59)	31	58.32
NO_2_ (µg/m^3^)	27.31 (23.75–31.00)	31	28.26	NO_2_ (µg/m^3^)	25.39 (21.55–29.37)	31	23.81
PM_2.5_ (µg/m^3^)	5.49 (4.67–6.30)	31	5.20	PM_2.5_ (µg/m^3^)	12.32 (9.40–15.57)	31	8.79
**Meteorological factors**				**Meteorological factors**			
T (°C)	8.19 (7.12–9.32)	31	8.00	T (°C)	8.34 (7.46–9.26)	31	7.90
RH (%)	61.45 (54.94–67.79)	31	59.00	RH (%)	68.08 (62.22–73.85)	31	73.80
WS (m/s)	4.49 (3.92–5.13)	31	4.40	WS (m/s)	1.45 (1.26–1.63)	31	1.54
**Total (March 2015–2020)**				**Total (March 2015–2020)**			
**Pollutant Concentrations**				**Pollutant Concentrations**			
CO (ppm) ^3,^*	0.30 (0.29–0.32)	186	0.27	CO (ppm) ^2,^**	0.21 (0.19–0.23)	186	0.19
O_3_ (µg/m^3^) ^3^	54.68 (52.68–56.73)	186	57.98	O_3_ (µg/m^3^) ^3,^*	53.57 (51.77–55.44)	186	54.47
NO_2_ (µg/m^3^) ^3,^*	34.45 (32.23–36.68)	186	31.78	NO_2_ (µg/m^3^) ^2,^**	39.06 (37.07–40.86)	186	40.26
PM_2.5_ (µg/m^3^) ^3,^*	6.96 (6.48–7.43)	186	5.93	PM_2.5_ (µg/m^3^) ^3^	16.35 (14.63–18.12)	186	11.10
**Meteorological factors**				**Meteorological factors**			
T (°C) ^3,^**	5.51 (4.86–6.19)	186	5.50	T (°C) ^3^	7.05 (6.43–7.60)	186	7.72
RH (%)	57.69 (55.31–60.37)	186	55.80	RH (%)	71.18 (69.70–72.65)	186	72.30
WS (m/s)	4.96 (4.71–5.23)	186	4.70	WS (m/s) ^3^	1.29 (1.22–1.36)	186	1.23

Notes: ^1^ a—no confinement (2015–2019) and b—confinement COVID-19 scenario (2020); ^2^ There are significantly distinct sets of means between years without march activity restriction (a, 2015–2019) and 2020 (b, coronavirus scenario), significantly different α = 0.05, Bonferroni test; ^3^ There are significant differences between some pairs of means and years, but there is no significantly distinct sets of means, a vs. b; ** analysis of variance *p* < 0.001, * analysis of variance *p* < 0.01, *p*–value represents results of the ANOVA procedure and compares the two groups of years, with (2020) and without (2015–2019) COVID-19 scenario.

**Table 3 ijerph-17-05067-t003:** Effects of activity restrictions and meteorology on air quality; predicting air quality during the month of March (2015–2020); measurements from São Paulo, Los Angeles, New York and Paris networks ^1,2^.

Dep. (unit)	CO (ppm)	O_3_ (µg/m^3^)	NO_2_ (µg/m^3^)	PM_2.5_ (µg/m^3^)
**São Paulo**				
**Coronavirus scenario**				
Activity restriction (a) ^3^	0.11 ± 0.001 *	−10.95 ± 0.002 *	4.00 ± 0.001 *	0.54 ± 0.000 *
**Meteorological factors**				
T (°C)	0.03 ± 0.000 *	3.49 ± 0.001 *	1.18 ± 0.000 *	0.74 ± 0.000 *
RH (%)	0.01 ± 0.000 *	−0.59 ± 0.000 *	−0.01 ± 0.000 *	−0.17 ± 0.000 *
WS (m/s)	−0.19 ± 0.001 *	0.66 ± 0.002 *	−10.40 ± 0.003 *	−5.49 ± 0.001 *
**Specific regressions**				
R^2^	0.64	0.50	0.45	0.53
Dep.	0.54	36.40	30.25	13.62
**Los Angeles**				
**Coronavirus scenario**				
Activity restriction (a) ^3^	0.09 ± 0.001 *	2.52 ± 0.02 *	11.01 ± 0.36 *	2.99 ± 0.001 *
**Meteorological factors**				
T (°C)	0.001 ± 0.000 *	2.54 ± 0.02 *	1.22 ± 0.000 *	0.66 ± 0.001 *
RH (%)	−0.003 ± 0.000 *	−0.05 ± 0.001 *	−0.33 ± 0.000 *	0.10 ± 0.000 *
WS (m/s)	−0.09 ± 0.001 *	0.08 ± 0.01 *	−8.31 ± 0.001 *	−1.66 ± 0.001 *
**Specific regressions**				
R^2^	0.58	0.25	0.55	0.37
Dep.	0.30	87.33	29.68	9.13
**New York**				
**Coronavirus scenario**				
Activity restriction (a) ^3^	0.11 ± 0.001 *	−7.96 ± 0.91 *	14.18 ± 0.004 *	2.03 ± 0.001 *
**Meteorological factors**				
T (°C)	0.004 ± 0.000 *	−0.01 ± 0.07	0.39 ± 0.001 *	0.12 ± 0.000 *
RH (%)	0.002 ± 0.000 *	−0.34 ± 0.02 *	0.20 ± 0.000 *	0.03 ± 0.000 *
WS (m/s)	−0.016 ± 0.000 *	3.22 ± 0.23 *	−3.38 ± 0.001 *	−0.23 ± 0.000 *
**Specific regressions**				
R^2^	0.23	0.32	0.23	0.14
Dep.	0.30	54.68	34.45	6.96
**Paris**				
**Coronavirus scenario**				
Activity restriction (a) ^3^	0.10 ± 0.001 *	−1.53 ± 0.001 *	12.52 ± 0.33 *	2.56 ± 0.23 *
**Meteorological factors**				
T (°C)	−0.003 ± 0.000 *	0.61 ± 0.000 *	0.09 ± 0.000 *	−0.47 ± 0.001 *
RH (%)	0.000 ± 0.000 *	−0.39 ± 0.000 *	0.04 ± 0.000 *	−0.18 ± 0.000 *
WS (m/s)	−0.18 ± 0.001 *	17.07 ± 0.000 *	−18.53 ± 0.000 *	−9.90 ± 0.003 *
**Specific regressions**				
R^2^	0.57	0.58	0.70	0.31
Dep.	0.21	53.57	39.06	16.34

Notes: activity restrictions due to coronavirus, dummy variable (a no restriction 5-y period, years 2015–2019); ^1^ the table report coefficients and standard errors for the São Paulo, Los Angeles, New York and Paris network station regressions; the number of regressors in the GLM was 4; ^2^ SE are calculated by bootstrapping (200 samples each); the pollutant and meteorology, clustering by day; in total 186 observations were sampled for each pollutant/city and the sample periods were March 2015–March 2020, including daily averages (Table 2a,b); ^3^ a, no activity restriction scenario, dummy variable; * analysis of variance *p* < 0.01; Model OLS (Ordinary least square). Dep. = dependent variable (Total (March 2015–2020)).

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
