# Peer review of "Air Quality during COVID-19 in Four Megacities: Lessons and Challenges for Public Health"

_ijerph, 2020, doi:10.3390/ijerph17145067_

Round 1

Reviewer 1 Report

Summary

The aim of this study is to determine how lower traffic levels associated with self distancing/isolation measures implemented to curb the spread of COVID-19 have affected ambient air quality levels in four large cities: Sao Paulo, Paris, Los Angeles and New York City. Concentrations of four pollutants mainly derived from vehicle emissions (CO, NO2, O3 and PM2.5) were compared for March 2020, to concentrations in March 2015-2019. It was found that average concentrations of CO, NO2 and PM2.5 were lower in March 2020 than in previous years, and that the difference for CO and NO2 was statistically significant in most cities. Ozone levels were higher in March 2020. The results give an indication of how air quality levels will be affected if policies are effectively implemented to reduce vehicle emissions in the cities.

Major Issues

I recommend that the following issues be addressed before this paper can be published:

  1. It has not been demonstrated that there was indeed less traffic in March 2020 than in preceeding years.

The underlying assumption in this study is that vehicle emissions were lower in March 2020 than in March of the previous five years, in the four cities included in the study. However, no data was presented to support this assumption. Indeed, the LDV and HDV activity data for Sao Paulo presented in Figure 7 shows that vehicle activity was higher for around two-thirds of March 2020. The Google COVID-19 Community Mobility Report cited considers transit stations, which are ‘places like public transport hubs such as subway, bus and train stations.’ The proportion of all vehicles that are public transport vehicles in the four cities is not mentioned. It is not clear how the number of people using public transport is linked to the total number of vehicles on the road.

  1. The period in early March before quarantine measures were implemented should not have been included in the ‘activity-restricted’ period

The Community Mobility data shows that mobility in transit stations reduced below the 3 January-6 February 2020 baseline on 9/10 March 2020 in LA and NYC, and on 15/16 March in Brazil and France. It is not clear, then, why the dates in early March were included in the ‘activity-restricted’ period. In Sao Paulo and Paris, CO concentrations in 2020 were below the 2015-2019 CO levels in early March, even when activity levels were not below normal, which suggests that other factors besides vehicles emissions are responsible for the lower CO levels in early March 2020.

  1. The main findings are not clearly presented

In the tables and discussion on the difference in the means between March 2020 and March 2015-2019 pollution levels, there is no clear statement on for which pollutants and cities there are statistically significant differences in the means. Moreover, Tables 2 and 3 need to be more comprehensively explained in the headings and notes.

  1. Statements about effect of lower pollution levels on health need to be more precise, and based on the analysis presented

Section 4.1 on the effects of the decrease in pollution concentrations on human health is completely speculative, and the effect of increased ozone concentrations on health was not weighed relative to the decreased concentrations of NO2 and PM2.5. I am of the opinion that this section adds little value to the paper. This paper shows a link between lower vehicle emissions and an improvement in CO and NO2 levels for two or three weeks. It is not clear that definitive statements on the effect of the lower CO and NO2 levels on human health can be made on the basis of such little data (and no information on health outcomes in this period).

Minor Issues

Please address the following:

  1. The title needs to be reworded to reflect that this study is about the effect of a reduction in transport activity on air quality. This study does not relate directly to the effects of COVID-19.

  1. The first sentence in the abstract needs to be restated, since the manuscript does not examine the quarantine policies themselves, but rather the effect of these policies on air pollution levels.

  1. Lines 60-74: It is not necessary to describe the WHO (2016) study in that much detail. Only the finding is relevant.

  1. Line 82: Correct formatting of the Wu et al. 2020 citation, and other similar citations throughout the manuscript.

  1. Line 82: Please confirm whether the Wu et al. 2020 study has now been published, and update accordingly.

  1. Section 2.1: While interesting, the descriptions of the study areas are lengthy and should be shortened significantly.

  1. Line 335: Please clarify what the ‘anthropogenic activity data’ is.

  1. Lines 419-428: Much of this should be moved to the section 2.3.

  1. Line 434: ‘Statistically lower’ should be rephrased to be more precise.

  1. Line 433: Figure 1 shows that the restrictions on economic activity only started later in March 2020 in all the cities (between 17 and 23 March). It is not clear why the entire month of March 2020 is considered to be a ‘period of restriction.’

  1. Lines 433-436: The finding here needs to be stated more precisely.

  1. Table 2: The presentation of this table can be improved:

  1. Note 1 links to specific places in the table, but I cannot find a link to Notes 2 and 3 in the table?
  2. What does ‘Obs’ denote?
  3. The p-value for the analysis of variance is indicated by means of asterisks after the value in the Obs column. What groups are compared for the analysis of variance in each case? It would be helpful if the p-value is stated in each case.

  1. Lines 577 to 578: The statement that ‘Anthropogenic activities, represented by the COVID-19 scenario, and meteorological variables were statistically significant for all the cities and pollutants’ is clumsily worded. Please improve.

  1. Line 589-590: Is the statement that ‘According to the COVID-19 Community Report [27], transport activities at 589 transit stations were reduced ~52% in São Paulo compared to ~78% in the other cities’ applicable to the March averages? Please clarify.

  1. Table 3: The table does not consider the effect of the coronavirus itself. Please change the table heading.

  1. Line 614: The statement that ‘Statistically significant differences between pollutants and meteorology were seen for all the pollutants’ is poorly expressed. Please restate.

  1. Figure 6 needs to be improved: the resolution is poor, and it is very difficult to discern from the legend in the line graphs which series is which.

  1. In Figure 7, it would be more appropriate to label the units on the axes rather than in the legend.

  1. Lines 701 to 703: The statement that ‘a minimization of health effects caused by reduced pollution levels could diminish pressure of health equipment and intensive care units which could be used for COVID-19 patients, thus helping to save lives’ needs to be further unpacked. The health effects of air pollution are usually considered in light of chronic effects, i.e. long-term effects of exposure to air pollution. This study focuses on mean monthly concentrations. However, the acute health effects of short-term exposure to high levels of air pollution are also relevant. It is not clear on which basis (reduction of chronic or acute health effects) the assertions here are made.

  1. Line 752-755: The WHO (2016) methodology is based on the relationship between annual average PM2.5 concentrations and health outcomes. It needs to be clarified whether the number of deaths calculated here are for one month, or for the whole year?

  1. Lines 851-852: The formulation of the statement ‘showing that air pollutants were improved at a higher rate through the restrictions, when compared to changes in meteorological parameters’ needs to be improved.

Reviewer 2 Report

This manuscript on lessons learned with respect to public health and air pollution in four megacities is very relevant and interesting. The choice of study areas is well argumentized, and the data applied for the analysis are appropriate and comprehensive.

In general, the manuscript is quite long and has several different stories to tell. It would be good to focus the manuscript more on one main story (the fact that activity changes means more than meteorological variations) and select minor stories that are relevant (e.g. the satellite imagery is not really supporting the main story with new evidence).

It would also be good to describe more clearly what are the main questions and main findings of the paper.

Another general comment regards the use of abbreviations: Please explain every abbreviation at first use (e.g. GLP and HDI-M are not explained) and be consistent with abbreviations and spelling (e.g. NY or NYC?, Los Angeles or LA?, sq mi or sq. mile?, corona or Corona etc).

There is substantial difference between the air pollutants in the study in terms of sources, components etc, the most apparent being that CO, NO2 and O3 are one-component gases, whereas PM2.5 consist of numerous compounds, some primarily emitted, some secondarily formed in the atmosphere, some inorganic and some organic. Secondary aerosols (term used by the authors) should be specified - is it the inorganic aerosols (where the toxicity is questioned in some studies) or the organic aerosols (where the biogenic emissions can play a tremendous role)? Please refer to literature concerning this matter (e.g. when it is addressed in line 857).

Specific comments:

62ff: The unit for concentrations is μg/m3 (not μg m3).

95: PM2.5 should be PM2.5

179-180: rephrase; PM2.5 is ALSO inhalable

181: the tendencies described - how are they related to meteorological variations?

238: a ; is missing before PM10

Figure 2-5: is this mean concentrations for all stations in each city?

Figure S2.4: UR should be RH

413: Secondary aerosols - specify (inorganic, organic or all?)

433ff: I don't understand this explanation, how does the overlapping confidence intervals prove statistical significance?

450: what can be possible explanations for the decrease in O3 in LA?

Table 2: please provide more explanatory text for this table

591ff: please explain what the numbers (%) means in terms of importance?

633: for ?? (something missing)

654: The atmosphere is highly non-linear, especially in urban environments, and the concentrations may therefore differ significantly (not slightly) from the emissions

671: the BC/CO2 data and traffic data should be introduced in the materials and methods section (but maybe consider if this side-story should be included in the manuscript at all?)

738-740: consider the other possible confounders between NO2 and Covid-19 counts; could it be that high NO2 levels is associated with high population density, and the high population density is the driving force?

753: these numbers appear very accurate, but in reality if it is 432 deaths or 501 is within the uncertainty of the estimate. Exposure-response functions for PM2.5 mostly focus on long-term exposure, and this study is focusing on a one-month emission change only. Please elaborate on the difference between long-term exposure and short-term exposure in relation to this finding

812-814:  Please provide a reference for this statement

Reviewer 3 Report

Thanks for the opportunity to revise the manuscript “Air Quality and COVID-19 in Megacities: Lessons Learned and Future Challenges for Public Health”. The manuscript had the goals to investigate the levels of four air pollutants during social distance policies implemented in four western megacities that have been highly affected by the COVID-10 pandemic and where air pollution is closely linked to motor vehicles.

This article has an extremely important topic as it shows how much human actions contribute to air pollution, and how much it affects human health. It is an extremely well written article, with clear methodology and very explicit results. In addition, it brings an important discussion about pollution, human action and health, which should always be discussed. I congratulate the authors for their excellent work.

Reviewer 4 Report

This is a very interesting, comprehensive and important paper and gives a valuable contribution to present discussions.

I have only a few comments:

line 51/52: " The health crisis caused by the virus is unprecedented":

Maybe it is worthwhile to mention at this point the so-called Spanish flue (the 1918 flu pandemic) which caused several million deaths.

line 79-82: Why is another typing format used here?

Figure 2-6: The figures look a little bit like bad screenshots and should be improved

Reviewer 5 Report

The paper is very well written and strongly substantiated. Reliable data sources were selected and used. The statistical and visual analysis performed is accurate and thorough.

I would recommend to the authors to compute the confidence intervals of the difference of means of air pollutants concentrations for scenarios (a) and (b) instead of subtracting the means of pollutant concentrations (lines 444-450), in order to have more robust statistical results. I would also suggest to use the confidence intervals for the mean value of each pollutant concentration instead of (Mean-sd, mean+sd) in the second column of table 2.

The column Med. in Table 2 stands for Median? I did not notice its use in the text. 

I had difficulties in understanding the "Obs" column in Table 2, containing "2" and "3" entries. If possible, ANOVA results should be reported in a more comprehensive way. 

All aspects of the issue studied are presented and analyzed.   

Reviewer 6 Report

This paper deals with the comparison of air quality in four big cities of the world during the shutdown and social distancing.

This article is interesting and it is related with a very current topic. It is well written and shows clearly the behavior of air quality in 4 major cities in the world during the pandemic. I consider that it should be published as it is a good contribution on the relationship of air quality with traffic and other activities decreasing, although I suggest some corrections:

  1. The paragraph in Page 2 Lines 123-128 with the results of this study should not be in the introduction but in the abstract or conclusions.
  2. It is understandable that the information of the different cities is not completely homogeneous. However, since the selection of cities with high traffic is vindicated in this article, therefore the concentrations of pollutants decreased, it is important that at least the contribution of vehicular traffic be presented for the 4 cities according to the emissions inventory, sinceit is only presented for Sao Paulo and Paris.
  3. The GLM model is not well explained. Please include in supplemental material a Table of P (dummy variable), a0, a1 and a2 and explain how was obtained the W in each case.
  4. The discussion section, that it is a very important section, should be related with their main findings such as pollutants decreasing or increasing, which they discussed widely in the results section. In opposite almost all discussion section is dedicated to possible human health consequences, but the authors did not make estimations or studies about that is issue. In my opinion, the inclusion of Zhu et al 2020 study, that is out of context in the case of NOx and PM, is unnecessary; far from providing important data to the manuscript, it diverts attention from the valuable data obtained in air quality. Authors noted that since in the conclusions there is nothing from the discussion section related to health. The relationship of pollutants with health is complex since can be synergistic effects during pollutants increasing or decreasing in different proportions. The general suggestions at the beginning of the discussion section and those of CO are good since they are related to the possible benefits in reducing somespecies and the increase in others. That is enough and round off the results obtained very well. My suggestion is to narrow down that discussion especially in phrases such as: “In relation to NO2, their results showed that a 10 μg/m3 increase in NO2 concentration was associated with a 6.9% increase in daily counts of COVID-19 cases”, as well as that related with PM.

Round 2

Reviewer 1 Report

Thank you for engaging so constructively with my comments.